# The Role and Function of Autobiographical Memory Narratives during the Emotional Processing of Breast Cancer Treatment: An Empirically-Derived Memory Coding System

**DOI:** 10.3390/ijerph20021492

**Published:** 2023-01-13

**Authors:** Maria Luisa Martino, Daniela Lemmo, Joshua Moylan, Caroline Stevenson, Laura Bonalume, Maria Francesca Freda, Jefferson A. Singer

**Affiliations:** 1Department of Humanities, Federico II University, 80133 Naples, Italy; 2Department of Psychology, Connecticut College, New London, CT 06107, USA; 3Department of Clinical Psychology (U.O.S.D), Territorial Healthcare Company, 20873 Brianza, Italy

**Keywords:** autobiographical narratives, memory, breast cancer, adaptation and coping, integration

## Abstract

Breast cancer (BC) in younger age is a critical and potentially traumatic experience that can interrupt the continuity of self-narrative during a crucial phase. In the Narrative Identity framework the translation of memories into autobiographical narratives is an internal and external process that plays a key role in meaning-making, social relationships and self-coherence. The aim of this study is to examine the role and function that autobiographical memory narratives (AMN) play in the process of adaptation to BC medical treatment. Seventeen BC women below 50 years received prompts to provide autobiographical memory narratives at four phases during their treatment (pre-hospitalization-T1-post-surgery-T2-chemo-radio therapy-T3-follow-up-T4). The Emotional Processing Scale (EPS) was also administered. In all, 68 AMN were collected. A three step procedure of data analysis was conducted. The first one, an empirically-derived memory coding manual to analyze key dimensions of AMN was developed: Agency; Emotional Regulation and Interpersonal Relations. Findings show a particular vulnerability in narrative identity faced by BC women during the shift from T1-T3. In the second one, an emotional coping profile for each woman focusing on the shift from T1-T3 was created. For the third step, these profiles were compared with the EPS scores. The final results suggest the capacity of the AMNs to differentiate the women’s emotional adaptation over the course of the BC treatment. Despite the study’s limitations, it supports the use of AMN as clinical device to construct a deeper knowledge and profiling trajectory of how women have internalized and elaborated past encounters with illness and help providers, as well as their prior experience of bodily/psychological health and integrity. This information adds to an understanding of their current efforts at recovery and adaptation. In this way we believe that the recollection of narrative memories, not only at the end of the cancer treatment but also during its process, could help the women to mend the broken continuity of their narrative self, as they seek to maintain a healthy balance of internal resources across their past, present, and projected future.

## 1. Introduction

### 1.1. Breast Cancer Experience in Young Age: The Psychological Impact

The onset of breast cancer (BC) in women under the age of 50 is a critical and potentially traumatic experience that can upset a woman’s life during a crucial phase of her lifespan and the achievement of her personal goals [1,2]. BC is the most common form of cancer in women and the highest incidence is in the 34- to 49-years age group (WHO), with an 87% survival rate. Despite the increasing number of women with BC under the age of 50, the psychological literature on this specific subject, although greatly required by vulnerable and at-risk women, still appears to be limited [3]. In our previous studies, we have shown that breast cancer for younger women increases clinical psychological risks: to feminine-specific concerns linked to the onset within an early phase of lifespan; to a search for resources between individual and relational aspects [1,4,5]. In addition, we demonstrated that psychological symptoms change during the different phases of medical treatment for BC. In particular, the levels of anxiety decrease from the phase of communication of the diagnosis to the chemo/radio therapy, while those of hostility increase. A crucial factor is the capacity of women to emotionally process the traumatic event. This capacity is a strong predictor of psychological symptoms during the phases of medical treatment [6,7,8,9].

The cancer can be interpreted as a stressor due to: the intangible and internal nature of the threat, the uncertainty about the disease outcome, the unpredictable trajectories, and the chronological aspects [10], which present recurrent stressors across different phases of the medical process [6,11]. These characteristics generate an accumulated burden of adversity, which may significantly affect later psychological functioning [12].

From a socio-constructivist and semiotic perspective of the mind [13,14,15,16,17], this event generates a sudden alteration of systems of meaning that support the relationship between the subject and the external world [18,19,20]. There is often a crisis of the sense of continuity of the self-identity [13,21,22]. This crisis is also expressed in an alteration of the temporal perspective that becomes characterized by a sense of fragmentation, a feeling of the suspension of life [22] and an uncertainty about the future [23]. This shift in the relationship to time produces an autobiographical discontinuity and it imposes a narrative urgency in the mind. This intense demand activates the search for a new synthesis of meanings through which to create a space between the pervasiveness of the emotions and the cancer-afflicted self, to give a new order and identifiable structure to these emotions, and ultimately to reorganize the meanings on which the coherence of one’s life story is based [24,25,26,27].

### 1.2. The Role of Narrative and Autobiographical Memory in Critical Experience

Within a Narrative Identity framework [28], autobiographical memory, the focus of the present study, is a particular narration of the self in which the narrator and the protagonist turn out to coincide. Narrative Identity develops and spans across the entire course of life. Individuals reconstruct the personal past, perceive the present, and anticipate the future in terms of an internalized and evolving self-story, which is an integrative narrative of the self that provides a sense of unity and purpose to an individual’s own life [29]. This process leads not only to the organization of events in a coherent way, but also to remembering these events from an integrative perspective. According to Smorti (2011), the relationship between narration and memory is close: personal narratives are comprised of memories, and memories must become narratives to be narrated. Autobiographical narrative is not a simple externalization of autobiographical memory because it implies that life events, maintained in memory, must be reported outside in the form of a story; this action of externalization entails an important transformation process because it provides a narrative structure to memory. Furthermore, the act of storytelling occurs in social relationships which are culturally established, so the shift from the autobiographical memory to the autobiographical narrative is a result from inside to outside (internal and external; cognitive and social) of the individual and they are constructed in the perspective of the present (hic et nunc) that is, in the perspective of the context of remembering and narrating. So, narratives are told to someone and/or to oneself. Consequently, they are even more constrained by the context of narrating and the emotional trigger that activates the process of retrieval and recollection [30,31]. Between memories and the narratives of memories, there is a circular and dynamic relationship of transformation of meanings over time [32,33,34] through the re-emergence of autobiographical material and its structuring into a narrative; people can benefit from cognitive and affective information, but also convert their memories into learning opportunities from experience, extrapolating from memory an integrative meaning that can represent a life lesson or insight [35,36,37].

Studies show that autobiographical narratives memories have a key role to contribute to successful problem solving, coping processes, and pursuing personal goals [38,39]. These kinds of memories serve important directive functions: they inform, guide, motivate, and inspire; they support social relationship and promote self-functions, such as self-continuity (diachronic integration), and they can sustain positive self-regard [40,41].

The process to translate cognitively processed information into “self-defining memories” [42] is our means of connecting specific past experiences to enduring concerns of the general personality system, as expressed through a coherent and continuous sense of narrative identity.

Autobiographical memories, recalled to the level of awareness, reflect the needs of the self. In fact, on the one hand they act as moderators of emotional processes, on the other they play an essential role in protecting and preserving a tolerable self-image. The relevance of some memories and certain objectives per se, with respect to others, derives from the specific responsiveness of the latter with respect to the evolutionary demands that arise in the various stages of life [35,36,37,43]. Specifically, the autobiographical narrative memory, in its constitution as a process, favors: the definition of the self, both in an absolute sense and in relation to others, and the emotional regulation of experiences, contributing, in the latter case, to the maintenance of psychological and physical well-being [34,44,45,46]. Some memories of events lived in one’s existential experience represent essential elements to define who we are; our self, in fact, is shaped by the way in which we manage to remember and reconstruct old experiences, as, by narrating our past, we also narrate about ourselves at the same time [44,45,46,47]. Through the narrative act, people try to weave a common thread between the different memories, which often can appear disconnected, so that they can be configured in a life story and, at the same time, outline a narrative identity; both the life story and the narrative identity contribute, therefore, to a sense of self in the world and in time, with respect to the past, and in the perspective of the future, to find a continuity even in moments of transition [42,44,45,46,47,48,49,50,51,52,53]. 

The construction of a narrative and the attribution of meaning to one’s experiences also play a fundamental role in the process of regulating emotions; many studies, in fact, have investigated the relationship that exists among narration, emotional regulation and autobiographical memory. The narratives, understood as emotionally connoted personal reconstructions, inevitably push people to interface with their emotions, in fact they require adjustment not only during the realization of an experience but also when it is recalled to consciousness and made an object of narration. Researchers have demonstrated that the meaning-making process in response to autobiographical memories is associated with better overall coping and adjustment, while memory content can provide a window into mood disturbances, including depression [35,42].

In this regard, some researchers have highlighted that integrating narratives to achieve a sense of coherence and continuity aids individuals in managing their emotional reactions and provides more control and predictability in their lives [44]. Another aspect closely associated with emotional regulation is the social sharing of emotions. Sharing experiences of affective intensity and strong impact with an interlocutor or more can help to modulate one’s emotions, since talking about one’s own feelings opens one up to the knowledge and awareness of the same elements necessary for the process of regulating emotions [33]. For all of these reasons, it would be of great value from a clinical standpoint to understand the role that memory recollections in the course of primary care consultations as well as diagnostic interviews might play in providing clinicians’ insight into patients’ coping patterns and emotion regulation.

Researchers have demonstrated that the meaning-making process in response to autobiographical memories is associated with better overall coping and adjustment, while memory content can provide a window into mood disturbances, including depression [35,42].

In this regard, some researchers have highlighted that the possibility, on the part of people, of intertwining narratives that convey a sense of coherence and continuity allows them to be able to better manage the emotional reactions associated with events and consequently perceive more control and predictability on one’s life [44]. Another aspect closely associated with emotional regulation is the social sharing of emotions. Sharing experiences of affective intensity and strong impact with an interlocutor or more can help to modulate one’s emotions, since talking about one’s own feelings opens one up to the knowledge and awareness of the same elements necessary for the process of regulating emotions [33]. For all of these reasons, it would be of great value from a clinical standpoint to understand the role that memory recollections in the course of primary care consultations as well as diagnostic interviews might play in providing clinicians’ insight into patients’ coping patterns and emotion regulation.

### 1.3. Autobiographical Memory Narrative and Breast Cancer Experience

Regarding the use of autobiographical memory narrative within the context of breast cancer, some studies attest that when people recall their past in association with life-threatening concerns or conflicts, the disease process can serve as a representative of these events. They have an integrative function because they contain lessons about the self or the world beyond the remembered events [54,55]. This integrated function may indicate that the individual engages in the construction of a life story and uses the past to inform their current experience of illness [56].

Despite the large number of studies in the field of autobiographical memory and the narrative of autobiographical memories, to date, few studies have explored the narrative functioning mediated by autobiographical memory during the course of medical treatment and the role played by them in the process of coping with the experience. Nieto (2019) suggested that intrusive memories of cancer experiences and avoidance related to such events are associated with autobiographical memory issues detected in depressive thinking.

Within the cancer context, studies have mainly focused on the long-survival phase and the construction and quality of an autobiographical memory of the illness experience following medical treatments. In particular, this research investigated the study of deficits in autobiographical memory due to medical treatments and/or psychological symptoms that accompany the traumatic impact of diagnosis and medical treatments.

Although little is known about possible deficits in autobiographical memory function in breast cancer survivors, depression is relatively common among these patients, as well as acute stress syndromes, and post-traumatic stress disorder (PTSD) [57,58,59]. The treatment’s adverse side effects can also induce a poor ability to recollect autobiographical memories [60,61]. In addition, some studies attest that psychopathological factors, such as depression and posttraumatic stress, appear to be more consistently associated with over-generality in autobiographical memory [62]. Moreover, the most anxious patients retrieved fewer emotional details for memories than the controls groups, and had lower self-representation scores than the least anxious patients, who had no deficits in emotional detail retrieval [63].

In one study, memories linked to the breast cancer experience reflected themes of “self-injurious memories”, even after they received successful treatment. Indeed, treatments themselves may bring about notable consequences for quality of life and self-perception. The “injured self” develops in breast cancer patients as illness representation play a crucial role in the consequences, perceptions, emotions, and adherence to treatments. Several daily difficulties could be associated with the Injured Self in terms of psychopathological symptoms and experience of self-fragmentation with negative influences on behaviors [64].

Starting from the aforementioned literature, the aim of this research is the exploration of the role and functions played by narratives of autobiographical memories during the different phases of medical treatment for young women with breast cancer. In particular, through an empirically derived coding system of autobiographical narrative memories, the ultimate goal was to highlight the power of autobiographical memory narratives to differentiate the level/capacity of the women’s emotional coping during the medical treatment. 

## 2. Materials and Methods

### 2.1. Participants and Recruitment 

The research was conducted at the National Cancer Institute Fondazione G. Pascale of Naples, the Italian national referral center for the treatment of neoplastic illnesses. The women who took part in this research were identified from medical reports and assessed for suitability in accordance with the following criteria: eligibility criteria were first admission to the hospital before the age of 50 and a diagnosis of infiltrating ductal BC;exclusion criteria were metastatic disease (stage IV), neoadjuvant therapy, and psychotherapeutic treatment in progress.

In the first phase of the research, 50 women were recruited at the pre-hospitalization phase. During the study, the total number of women undergoing the four longitudinal phases was 17 (see Table 1), all below 50 years of age (M = 44.47; SD = 3.87). The dropout of women from the first phase of recruitment was due to changes in the hospital structures, a worsening of the cancer condition, a desire on the part of the patients not to continue, and a lack of available time. The meetings took place by means of face-to-face interviews in an ad hoc room of the hospital. The data were collected during the year 2018.

### 2.2. Ethical Approval 

The research was conducted within the framework of the STAR Programme, financially supported by UniNA and Compagnia di San Paolo. The research was co-constructed in collaboration with the hospital’s psychology service and breast unit. The study was approved by the ‘Ethical Committee of The National Cancer Institute Pascale of Naples with managerial decision of N. 36 del 18/01/2018.’ The study respected the American Psychological Association’s Ethical Principles and Code of Conduct as well as the principles of the Declaration of Helsinki. The hospital’s psychology service provided a location and facilities for the monitoring meetings and the treatment of the women who wanted to continue receiving psychotherapeutic support over time. The participants were informed about the study’s aims and procedures and were assured that their participation was voluntary and that their responses would remain anonymous. The women volunteered to participate by providing an informed consent in a written form, with the hospital approving the privacy policy.

### 2.3. Procedure and Longitudinal Recruitment 

Women met during four different phases of the first year of medical treatment. Every medical phase, constituting a turning point of the medical treatment protocols, reflected turning points in the meaning of the woman’s relationship with BC over time and its psychic challenges [54]. 

In Phase I: facing the unknown, the woman is still undergoing diagnostic investigation of a suspected nodularity. 

In Phase II: the impact of the critical valence of the disease, the woman learned about the severity of her pathology (receiving histological examination), has undergone surgery for malignant nodularity, and decides the therapeutic path to be taken.

In Phase III: relationship with a changed body identity, the woman is faced with postoperative chemotherapy or radiotherapy treatments that affect her relationship with her body.

In Phase IV: the construction of a new continuity, the woman returns to the daily routine of life and integrates the maintenance phase, which will last for at least five years. The woman finds herself recovering spaces of autonomy and gradually reducing dependence on the medical institution since she only goes for follow-up.

### 2.4. EPS: Emotional Processing Scale 

EPS-25 is a self-report questionnaire that is designed to identify, quantify, and differentiate the types of EP styles and potential deficits in healthy individuals and those with psychological or physical disorders, as well as to measure the changes in EP as a result of therapy or interventions for physical or psychological disorders, and to assess the contribution of poor EP to the development of psychosomatic and psychological disorders. This scale comprises five subscales, each with five items that are rated on a 10-point (0–9) attitudinal scale: suppression (excessive control of emotional experiences and expressions); signs of unprocessed emotions (intrusive and persistent emotional experiences); unregulated emotions (inability to control one’s emotions); avoidance (avoidance of negative emotional triggers); and impoverished emotional experiences (detached experience of emotions due to poor emotional insights). The total EPS score is obtained by adding the scores of every item completed for the subscale and dividing by the number of items, to give a mean score in a range which goes from 0 to 9. A higher score indicates a poorer EP. The total scores based on a healthy comparison group are as follows: very low (1.1), low (1.7), average (between 2.0 and 5.2), high (5.6), and very high (6.1) [65]. These scores were converted to T scores and scores two deviations above the norm of 50 (70+) were considered high and very high scores, indicating greater dysfunction in emotional processing on each of the subscales.

### 2.5. Narrative Ad Hoc in Deep Interview

We constructed an original ad hoc narrative interview, named the Early Breast Cancer-Processing Trauma Interview (EBC-PTI), to explore young women’s narrative sense-making processes within the BC experience [66] in every phase of their therapeutic path. The same narrative interview during T1, T2, T3, and T4, involved nine open questions that started from the initial request to narrate the illness’s experience from the moment it appeared until the time of the interview. Each question was intended as a narrative prompt that would allow for the recounting of their experience and the opportunity to engage in meaning-making with regard to what they were undergoing. The interview was constructed to activate different forms of narrative discourse and to explore different areas of the experience: area 1, the story of the experience with a specific focus on the actual phase; area 2, attributions about the causes of cancer; area 3, an episodic deepening and examination of relationships with similar experiences—this was the area most likely to activate autobiographical memory narratives; area 4, the specific sense of crisis and change; and area 5, resources.

This study looks specifically at area 3—the prompt regarding of the exploration of episodic memory, and the recollection of autobiographical memories regarding similar past experiences.

The exact prompt for area 3 was: “Could you tell me if you have had other experiences in your life that you consider similar in any way, even if only emotionally, to this one? Could you tell me how you managed them?”

The interview was conducted in an ad hoc room of the hospital; it had an average duration of approximately 40 min and was recorded and then transcribed verbatim. The interview was conducted by two women psychologists who are experts in clinical psychology and narrative methodology. Matching the gender of the interviews and the participants was a conscious choice to promote the women’s open reflection and narration. The researchers were young women; this allowed them to develop an empathic exchange with the patients.

## 3. Data Analysis

### A Step by Step Empirically-Derived Procedure

Data Reduction and Translation

In order to examine the specific relationship of the autobiographical memory narratives to the women’s coping capacities across the 4 phases of treatment, the research team extracted these portions of the interviews from the larger interview protocols for each woman. The resulting transcripts contained only the women’s responses to the area 3 prompt for each phase (68 autobiographical memory narratives). Since the autobiographical memory researchers within the research team were English speakers, the first step was to translate the memory narrative excerpts into English. In order to ensure that the transcripts could preserve their meaning after translation and at the same time remain accurate to the original Italian, the American researchers identified two advanced undergraduate students at Connecticut College who were majoring both in Psychology and Italian Studies. After these assistants had translated the memory narrative excerpts to English, the Italian researchers then back-translated a portion of the excerpts into Italian to make sure that the English versions were true to the original Italian. The translations were highly accurate, but any small discrepancies and confusions were clarified through discussion until a full consensus was reached on the veracity of the translations.

2.Development of a Memory Coding Manual System

Although the research team ultimately determined that, with the size of the sample and variation across phases in participants’ responses (both in length and absence of response), a qualitative analysis rather than a quantitative analysis would be most appropriate, the initial procedure was to develop a coding manual that would allow for the assignment of rating values for different variables related to coping and emotional processing.

To develop this manual, the English-speaking research team engaged in a thematic analysis [67] of the 17 participants’ four phases of memory narrative excerpts. The head of the American research team, a faculty member with expertise in qualitative analysis and autobiographical memory (JAS), read through the 17 memory narrative protocols, seeking to identify repetitions and salient coping and affective responses that would yield initial codes. After grouping these codes, three themes emerged: agency; emotion regulation; and interpersonal relations (Table 2). Words and phrases associated with each of these themes can be found in Appendix A. For example, low agency was associated with participants using the word, “helpless”, poor emotion regulation was cued by words like “traumatized”, and low interpersonal support was evidenced by phrases like “I faced it alone”. 

The American researcher then asked one of the undergraduate research assistants (CS) to determine the applicability of these themes by independently coding the memories. After confirmation from the research assistant, these themes were shared with the Italian research group who similarly confirmed the comprehensiveness of these three major themes. On the basis of this consensus, the American team of the faculty member and two research assistants developed The Breast Cancer Survivors’ Autobiographical Memory Narratives Coding Manual (see Appendix A).

Two undergraduate research assistants (one from the original team (CS) and a new second undergraduate assistant (JM)) then used the manual to code and rate all 17 participants’ memory narratives across the four phases of the disease and treatment course. Since the most robust and detailed memory narratives emerged in the first phase (diagnosis) and the third phase (post-surgery and at commencement of chemo and/or radiation treatment), the two assistants paid particularly close attention to the memory narratives from these two phases. Using the coding results, the second research assistant (JM) created 17 emotion coping/processing profiles, synthesizing the agency, emotion regulation, and interpersonal relations findings (Appendix B for the 17 personality profiles). It is important to highlight that these profiles were based solely on the memory narrative data with no other knowledge of the participants and/or their full interview protocols.

## 4. Results

### 4.1. Dividing the Emotion Coping/Profiles

Once the profiles were generated, the faculty member (JAS) and research assistant (JM) divided the 17 participants into two groups on the basis of their profiles—healthier emotional processing and less healthy emotional processing and coping. We made the decision to create the two groups in order to examine how the groups might differ in their scores on the EPS. This would provide us with a pilot approach for assessing the validity of our memory coding system. 

In order to create the two groups, JM conducted a holistic re-reading of each participant’s memory protocols and also reviewed the rating scores on the three dimensions of agency, emotion regulation, and interpersonal support. This review led to the assignment of the 10 participants who showed consistent high levels of struggle across the three dimensions to the less healthy coping group and seven participants to the more healthy coping group. JAS then conducted an independent review of the memory protocols and rating scores; this review confirmed the two groups as being appropriately assigned.

Individuals in the healthier coping group showed more evidence of positive agentic coping (greater confidence in overcoming the disease and its aftermath), more effective emotional regulation (greater optimism and use of more positive coping strategies), and less interpersonal distress (greater connection with others either through receiving/giving support and/or trusting in the medical team and/or their faith). 

### 4.2. Validating the Emotion Coping/Processing Profiles

As the final step in this pilot process, the research team examined the Phase 1 and Phase 3 self-report scores on the five subscales of the EPS for the two groups created from the analysis of the autobiographical memory narrative excerpts. Given the multiple subscale ratings across the phases, the team asked the basic question: in comparing the two groups, what would be the percentage of highly problematic emotional processing scores (defined as two standard deviations above the norm) for the healthier vs. the less healthy emotion coping/processing groups? 

The results indicated that the healthier group had 23% of their responses in the problematic range, while the less healthy group had 40%. Figure 1, Figure 2, Figure 3, Figure 4 and Figure 5 display the contrast between the two groups (see Figure 1, Figure 2, Figure 3, Figure 4 and Figure 5). In particular, one can see stark differences in Phase 3 on the suppression and unprocessed emotions subscales. There is also a marked difference summed across the phases for the controllability of emotions. Finally, one can see a much greater awareness of their emotions in Phase 3 for the healthier group. As an additional quantitative analysis, Table 3 depicts the means and standard deviations on the EPS subscales, divided by the healthier and less healthy coping groups. Given the small n’s, we chose to conduct a sign test on the 10 mean comparisons. Nine of the 10 means for the less healthy coping group were higher than the means for the more healthy coping group, indicating greater difficulty in emotional processing, *z =* 2.53, *p =* 0.011 (see Table 3). 

### 4.3. Summary of the Findings

Drawing on previous clinical studies of autobiographical memory narratives’ predictive power [38], the current study examined the heuristic ability of memory narrative excerpts to provide meaningful insight into patients’ personality and coping dynamics. A team of memory researchers isolated and coded 17 breast cancer survivors’ memory narratives taken from extensive interviews over the course of their diagnosis and treatment. With no knowledge of the content of the remainder of the interviews or the specific demographics or details of the 17 survivors, they generated profiles of each survivor by coding their agency, emotional regulation, and interpersonal interactions, as depicted in their autobiographical memory narrative excerpts.

These profiles were then divided into two groups, differentiated by a holistic judgment of the degree of healthy responding to their disease experience. Healthier responses were differentiated by greater expression of positive agency, honest but ultimately optimistic emotional coping, and more adaptive interpersonal functioning (engagement with others and higher degree of trust and faith). To test the validity of this division into two groups, the researchers compared the groups’ self-reported emotional processing scores on the five subscales of the Emotional Processing Scale (EPS). The group defined as less healthy had 40% of their EPS subscale scores at or above 2 standard deviations from the normative scores (indicating less effective emotional processing), while only 23% of the healthier group were at that level. A comparison of means on the subscales for the two groups was also significant. These findings confirmed the effectiveness of the memory coding analysis in differentiating patterns of more and less effective emotional coping styles (see Table 3, Figure 1, Figure 2, Figure 3, Figure 4 and Figure 5)

## 5. Discussion

We find it interesting to focus attention on the presence/absence of some emotional defensive styles in the relationship between EPS and groups of women (healthy and less healthy). Specifically, we refer to the two styles of emotional processing such as: suppression and unprocessed emotion in the less healthy emotion/coping processing group. These defensive styles of emotional processing could represent a block in the access and emotional contact with the current experience of illness that prevents the process of recovering those internalized and integrated autobiographical memories of one’s complex past experiences that can serve as support to the needs of actual self.

On the other hand, the greater awareness and integration of one’s own emotional style of relationship with the experience of illness and treatment in women “more healthy” (this is shown by the total absence of controllability and unprocessed emotions) could indicate that the greater softening of defenses allows access to those elaborated and integrated memories capable of sustaining the current resources and needs of the self. 

The results are further preliminary evidence that autobiographical memory extracts from clinical data can provide valuable insight into individuals’ functioning (agency; emotion regulation; interpersonal relations) and overall coping mechanisms and defenses. This result argues for continued attention to the assessment of memory data from clients in clinical interviews and ongoing clinical interactions. Autobiographical memories offer a window into the overall narrative identity of the individual by revealing the continuity of experience over the life course. A knowledge of how individuals have internalized past encounters with illness and help providers, as well as their prior experience of bodily/psychological health and integrity, may deepen our understanding of their current efforts at recovery and their capacity for resilience.

In conclusion, the analysis of memory data provides an indirect way of ascertaining clients’ psychological health beyond self-report measures of well-being and coping. Although correlated with self-report data, narratives offer a more nuanced and diachronic understanding of coping dynamics—an understanding that helps one to see fluctuations back and forth in coping capacity and strategy.

## 6. Conclusions

The current study is by no means definitive or prescriptive regarding what optimal emotional processing and coping might be in surviving breast cancer and/or serious illness. More modestly, it is a preliminary step in demonstrating the viability and practicality of attending to memory narrative data as a source of meaningful insight in working with survivors over time. Future research would benefit from improved memory narrative measures and continued refinement of techniques of solicitation, extraction, data analysis, and interpretation. 

### Clinical Implications

The current study reinforces the value of autobiographical memory narrative analysis in clinical health, and personality psychology. In particular, the use of narrative memories, collected through narrative interviews during the first phase of illness experience, can offer insight into how these individuals have internalized past encounters with illness and how this internalization reflects their sense of bodily integrity, psychological health, and their relationship to health providers. This knowledge may help providers forecast psychological responses to treatment and aid them in assisting the women in managing and adapting to their breast cancer experience. The clinical health psychology team can plan psychological support consistent with the profiles that emerge from the narrative memories as a means of mitigating some of the coping challenges that are raised over the phases of the cancer treatment. The incorporation of memory narrative assessment into the treatment is another extension of the clinical psychologist’s ability to “observe and listen in depth” to the woman, enhancing holistic treatment that goes beyond the simple mechanics of the diagnosis, surgery, and subsequent adjunctive treatment with radiation and/or chemotherapy.

In addition, clinical engagement with autobiographical memory narratives is also useful at the end of cancer treatment and in the follow-up phase. A knowledge of the survivor’s previous narrative identity structure and themes enables the treatment team to support her efforts to address the discontinuities in her narrative caused by the illness and to rebuild a sense of coherence and sustained meaning, despite an altered sense of self.

### Limitations

There are also several limitations to this preliminary study. First, it was based on a small number of participants, 17 in total, which limited efforts at meaningful inferential statistical analyses. Second, since the original sample was 50 in total, one must be cautious in making generalizations with regard to the 17 participants who persisted across the full study’s four phases. Third, the memory researchers joined this project after the original data had been collected and did not have the opportunity to provide input on the wording of the autobiographical memory narrative prompt. The wording was: “Could you tell me if you have had other experiences in your life that you consider similar in any way, even if only emotionally, to this one? Could you tell me how you managed them?” In future studies, we might suggest using the following wording: “Could you please share a memory from your own life that you might consider similar to what you are undergoing at present? In recalling the memory, please give us some sense of what you were experiencing emotionally at that time, how you felt about yourself, and the role that others in your life played?” This wording, or something similar, might engage the participant to provide a more specific episodic memory with greater detail and imagistic quality. Fourth, the thematic coding of agency, emotional regulation, and interpersonal reactions were inductively developed from the memory narrative protocols of these 17 participants. If the researchers had applied an a priori theoretical lens or had developed the coding manual from a larger sample, a larger number of categories might have emerged along with different emphases in those categories.

## Figures and Tables

**Figure 1 ijerph-20-01492-f001:**
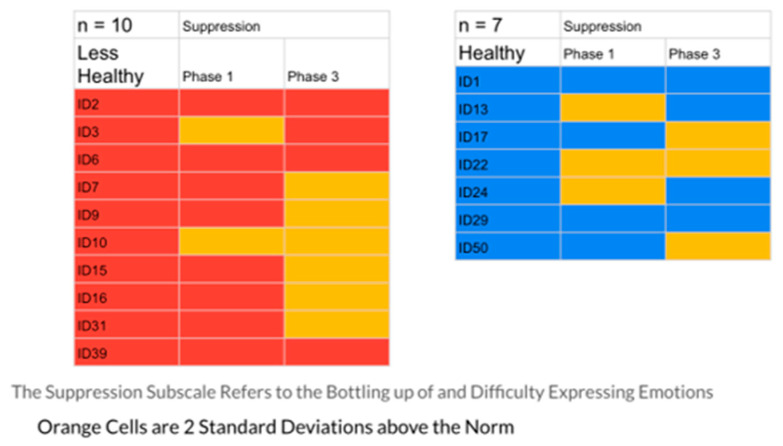
The Suppression Scale EPS: Less Healthy vs. Healthy Coping Groups.

**Figure 2 ijerph-20-01492-f002:**
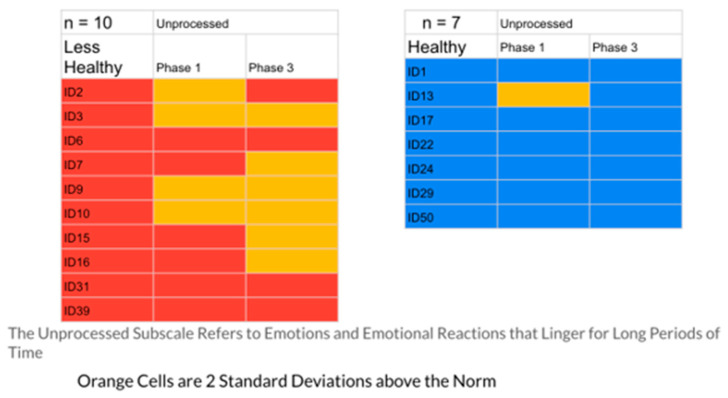
The Unprocessed Scale EPS: Less Healthy vs. Healthy Coping Groups.

**Figure 3 ijerph-20-01492-f003:**
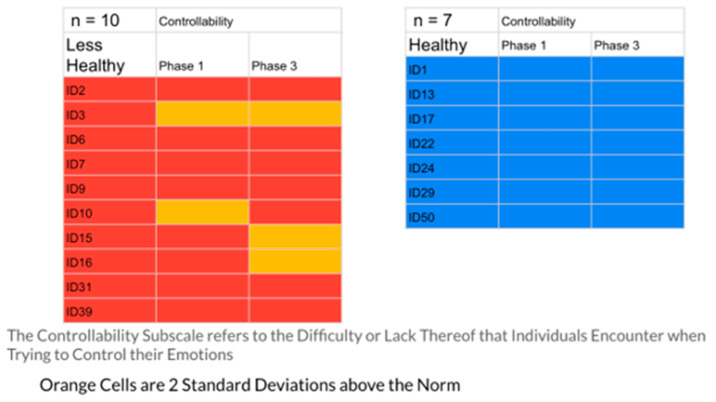
The Controllability Scale EPS: Less Healthy vs. Healthy Coping Groups.

**Figure 4 ijerph-20-01492-f004:**
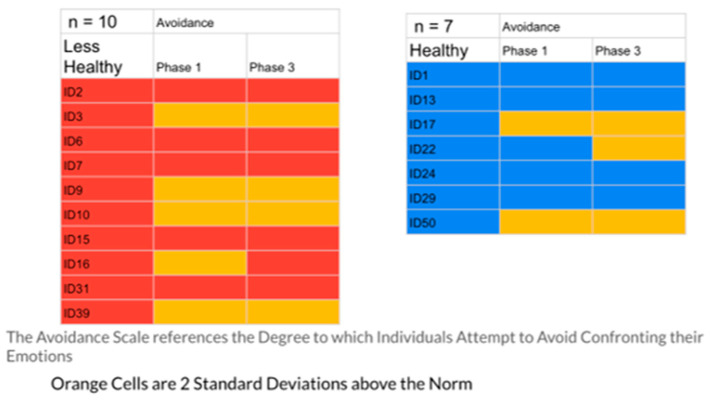
The Avoidance Scale EPS: Less Healthy vs. Healthy Coping Groups.

**Figure 5 ijerph-20-01492-f005:**
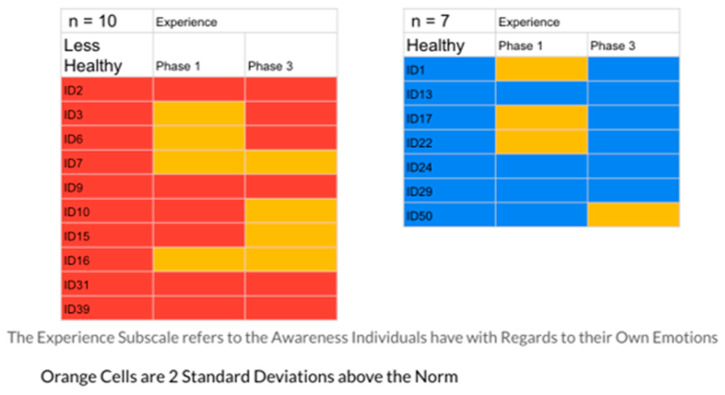
The Experience Scale EPS: Less Healthy vs. Healthy Coping Groups.

**Table 1 ijerph-20-01492-t001:** Sociodemographic characteristics of the breast cancer women participated in the study.

Id	Age	Educational Level	Job Position	Marital Status	Number of Children	Age of Children
1	48	Elementary/secondary	Housewife	Married	2	23; 16
2	45	Elementary/secondary	Housewife	Married	2	20; 9
3	39	High school	Housewife	Married	1	7
6	45	High school	Employee	Single	0	0
7	49	Elementary/secondary	Housewife	Married	1	25
9	36	High school	Employee	Married	1	8
10	46	Elementary/secondary	Housewife	Married	2	32; 26
13	47	Elementary/secondary	Freelance	Married	2	21; 11
15	49	Elementary/secondary	Freelance	Married	1	11
16	44	Elementary/secondary	Housewife	Married	3	15; 13; 9
17	41	High school	Employee	Single	2	4; 2
22	42	Elementary/secondary	Employee	Single	3	22; 20; 18
24	48	Elementary/secondary	Housewife	Married	2	19; 14
29	46	High school	Housewife	Single	0	0
31	48	Degree	Employee	Married	2	11; 13
39	44	High school	Employee	Separated	1	22
50	39	High school	Employee	Married	1	9

**Table 2 ijerph-20-01492-t002:** Codes of manual system.

Agency	The degree to which the participants felt they could make an active response or have any sense of control in response to their diagnosis, disease process, and treatment.
Emotion Regulation	Encompassed their range of coping responses at each phase of their cancer diagnosis and treatment; these responses included acceptance; distraction/denial; distress/hopelessness; and active positive coping.
Interpersonal Relations	Included accepting support; supporting others; distancing/self-reliance; expressing trust/faith in a greater power, whether God or the medical team; and interpersonal distress caused by negative family and/or other interactions.

**Table 3 ijerph-20-01492-t003:** Means and standard deviations for the emotional processing subscales.

Divided by Less Healthy and More Coping Styles Based on Memory Coding				
	Less healthy		More healthy	
	Coping group		Coping group	
	(n = 10)		(n = 7)	
	Phase 1	Phase 3	Phase 1	Phase 3
	Mean (S.D.)	Mean (S.D.)	Mean (S.D.)	Mean (S.D.)
Suppression	4.27 (2.17)	6.20 (2.61)	5.54 (1.94)	4.20 (2.98)
Unprocessed	4.94 (2.79)	5.80 (1.75)	3.83 (2.72)	2.56 (2.00)
Controllability	3.21 (2.46)	4.53 (1.96)	1.37 (1.24)	2.16 (1.62)
Avoidance	4.58 (2.96)	5.20 (2.32)	4.06 (1.26)	4.68 (2.82)
Experience	3.81 (1.62)	3.91 (1.92)	3.33 (1.70)	3.12 (2.21)

## Data Availability

The data presented in this study are available on request from the corresponding author. The data are not publicly available due to the protection of the privacy of the women involved.

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
