# Peer review of "The Role and Function of Autobiographical Memory Narratives during the Emotional Processing of Breast Cancer Treatment: An Empirically-Derived Memory Coding System"

_ijerph, 2023, doi:10.3390/ijerph20021492_

Round 1

Reviewer 1 Report

Although the overwriting of the manuscript is almost ready for publication, I have some questions that might need author's explanation:

1. Conclusion

Please explore on why and how the current study 'reinforces the value of autobiographical memory narrative analysis in clinical, health, and personality psychology'? Please state specifically the clinical implications, possible impact on public health, or information on personal psychology, point by point regarding these different aspect.

2. Similarly, the fourth paragraph of the Discussion section: why does this study 'provide valuable insight into individuals' functioning'?

3. Also, why is this study may be relevant to the scope of this magazine? As noted from the magazine's website: 'International Journal of Environmental Research and Public Health  covers Environmental Sciences and Engineering, Public Health, Environmental Health, Occupational Hygiene, Health Economic and Global Health Research, etc.  ' What are the points that the study was related to public health, or global health research? these may also be relevant to authors explanations for my Question #1.

Author Response

Reviewer 1 comments

First, thank you for the extremely helpful comments. 

Although the overwriting of the manuscript is almost ready for publication, I have some questions that might need author's explanation:

  1. Conclusion

Please explore on why and how the current study 'reinforces the value of autobiographical memory narrative analysis in clinical, health, and personality psychology'? Please state specifically the clinical implications, possible impact on public health, or information on personal psychology, point by point regarding these different aspect.

  1. Similarly, the fourth paragraph of the Discussion section: why does this study 'provide valuable insight into individuals' functioning'?

Regarding points 1 and 2, we have now inserted the following paragraph in the Discussion:

In particular, the use of narrative memories, collected through narrative interviews during the first phase of illness experience, can offer insight into how these individuals have internalized past encounters with illness and how this internalization reflects their sense of bodily integrity, psychological health, and their relationship to health providers.  This knowledge may help providers forecast psychological responses to treatment and aid them in assisting the women in managing and adapting to their breast cancer experience.  The clinical health psychology team can plan psychological support consistent with the profiles that emerge from the narrative memories as a means of mitigating some of the coping challenges that are raised over the phases of the cancer treatment. The incorporation of memory narrative assessment into the treatment is another extension of the clinical psychologist's ability to "observe and listen in depth" to the woman, enhancing holistic treatment that goes beyond the simple mechanics of the diagnosis, surgery, and subsequent adjunctive treatment with radiation and/or chemotherapy. 

In addition, clinical engagement with autobiographical memory narratives is also useful at the end of cancer treatment and in the follow-up phase. Knowledge of the survivor’s previous narrative identity structure and themes enables the treatment team to support her efforts to address the discontinuities in her narrative caused by the illness and to rebuild a sense of coherence and sustained meaning, despite an altered sense of self.

  1. Also, why is this study may be relevant to the scope of this magazine? As noted from the magazine's website: 'International Journal of Environmental Research and Public Healthcovers Environmental Sciences and Engineering, Public Health, Environmental Health, Occupational Hygiene, Health Economic and Global Health Research, etc. ' What are the points that the study was related to public health, or global health research? these may also be relevant to authors explanations for my Question #1.

Thank you for raising this question.  The article was submitted into the special issue “Clinical Health Psychology for Cancer Experience: Current Challenges among Prevention, Treatment and Chronicity” of this journal.  Please see the call for submissions and description at

https://www.mdpi.com/journal/ijerph/special_issues/clinical_health_psychology_cancerT

The article fits very well with the aim of the special issue call and it fits with the Public Health aim of the Journal.  Please see the updated conclusion section.

Reviewer 2 Report

This manuscript was prepared in a good manner. However, why you divided samples in to groups? If you could report EPS as a numerical value (Mean and standard deviation), I think the results section had been improved.

Author Response

Reviewer 2 Comments

This manuscript was prepared in a good manner. However, why you divided samples in to groups? If you could report EPS as a numerical value (Mean and standard deviation), I think the results section had been improved.

We explain why we divided the groups.  We have also now added a Table of the means and standard deviations for the EPS subscales (see Table 3 in the text) and demonstrated that they significantly differ between the two groups, the less healthy and more healthy coping style groups.   Here is the text introducing this table and accompanying analysis:

As an additional quantitative analysis, Table 3 depicts the means and standard deviations on the EPS subscales, divided by the healthier and less healthy coping groups. Given the small n’s, we chose to conduct a sign test on the 10 mean comparisons.  Nine of the 10 means for the less healthy coping group were higher than the means for the more healthy coping group, indicating greater difficulty in emotional processing, z = 2.53,
